

# Self-care practices and associated factors among type 2 diabetes mellitus patients attending public hospitals in Bale zone, Oromia region, Ethiopia

Eshetu Zemen[1], Yimer Seid Yimer[1], Negussie Deyessa Kabeta[1] and Yonas Abebe[2]

[1] Department of Epidemiology and Biostatistics, School of Public Health, Addis Ababa University, Addis Ababa, Ethiopia
[2] Addis Ababa University, Addis Ababa, Ethiopia

Corresponding author
Eshetu Zemen,
zemeneshetu5@gmail.com

## ABSTRACT

**Background:** Type 2 diabetes is a chronic condition characterized by elevated blood sugar levels. Effective self-care, including medication management, dietary changes, exercise, regular blood sugar monitoring, education and support, stress management, regular healthcare visits, foot care, and sleep hygiene, is essential for its management.

**Objective:** The study aimed to gather data on how well patients managed their diabetes through self-care activities, which are crucial for maintaining optimal health outcomes.

**Methods:** A cross-sectional study was conducted in public hospitals in the Bale zone of Ethiopia from February 5 to March 22, 2024. Using systematic random sampling, 411 patients over 18 with type 2 diabetes were selected. Data were collected *via* interviewer-administered questionnaires, entered into Kobotoolbox, and analyzed using SPSS version 26. Choosing all existing public hospitals in Bale Zone provided comprehensive insights into type 2 diabetes management while adhering to ethical standards, ensuring participant protection and enhanced research credibility through valid instruments designed for accurate data collection.

**Results:** In this study, 59.4% of the 411 participants demonstrated satisfactory diabetes self-care practices, while 40.6% exhibited inadequate practices. Significant factors influencing self-care included higher income (adjusted odds ratio (AOR): 2.39, 95% confidence interval (CI) [1.19–4.80], $P < 0.014$), private sector employment (AOR: 2.09, 95% CI [1.06–4.13], $P < 0.033$), receiving diabetic education (AOR: 2.85, 95% CI [1.33–6.12], $P < 0.007$), membership in a diabetic association (AOR: 1.85, 95% CI [0.93–3.67], $P < 0.077$), possessing good self-care knowledge (AOR: 2.04, 95% CI [1.24–3.34], $P < 0.004$), and having no diabetic complications (AOR: 2.68, 95% CI [1.64–4.36], $P < 0.000$).

**Conclusion:** Diabetes self-care practices among type 2 diabetes patients in Bale, Ethiopia, are not sufficient. These practices are affected by several factors, such as socioeconomic status, access to diabetes education, membership in diabetes associations, and overall knowledge about self-care. Targeted support and education are crucial for individuals with lower incomes and those in non-private jobs. Enhancing access to diabetic education and promoting membership in diabetic associations can significantly improve self-care practices. Furthermore, it is essential

to focus on knowledge enhancement and preventive care for complications during public hospital follow-ups in the Bale Zone.

## INTRODUCTION

Diabetes mellitus is a chronic disease that occurs when the pancreas cannot produce enough insulin or the body cannot use insulin effectively. The incidence and consequences of diabetes have increased steadily over the past decade (*Chan, 2016*). The *American Diabetes Association (2020)* identifies several forms of diabetes, including type 1, type 2, gestational diabetes, and other related medical conditions. Type 1 refers to the process of beta cell degeneration that can lead to diabetes, where insulin is required for survival to avoid ketoacidosis, coma, and death. Pregnancy is the period when diabetes occurs (*American Diabetes Association, 2021*). Type 2 diabetes is caused by a progressive increase in the amount of insulin released by B cells, often accompanied by insulin resistance (*American Diabetes Association, 2021*). Type 2 diabetes can be caused by many factors; urbanization, change in diet, decrease in physical activity, obesity, ageing, family history of diabetes, belonging to certain groups, and lifestyle changes are some of these factors (*Söderhamn, 2000*; *Baine & Rasheed, 1979*).

The main goal in diabetes treatment is to keep the blood sugar level between 70–120 mg/dl before meals and below 140 mg/dl 2 h after meals (*Kassahun, Eshetie & Gesesew, 2016*).

The American Diabetes Association recommends self-monitoring of blood sugar (SMBG) for people with diabetes (*American Diabetes Association, 2021*). Diabetes should be treated primarily with non-drug treatments, including lifestyle changes. People with type 2 diabetes can achieve and maintain adequate blood sugar levels through a healthy lifestyle that includes a healthy diet and regular exercise (*Abdu et al., 2023*). Dietary patterns represent the best way to measure dietary intake (*Desa et al., 2020*; *Dere et al., 2016*). They have played an important role in helping to clarify the relationship between diet and chronic diseases in recent years (*Pawlak, 2017*; *Jannasch, Kröger & Schulze, 2017*; *Esposito et al., 2015*). The number of patients continues to increase in Ethiopia (*Dere et al., 2016*; *Adem et al., 2020*; *Ambaw et al., 2021*). Therefore, in this study, we aimed to evaluate diabetes self-care and changes in patients with type 2 diabetes in a public hospital in Bale Region, Ethiopia.

Regarding the non-communicable disease (NCD) targets linked to diet, Ethiopia has achieved a little headway (*Seifu et al., 2021*). According to estimates, diabetes affects 5.8% of men and 5.0% of adult women (*Chinnappan et al., 2020*). Diabetes is evolving into a severe public health issue, necessitating ongoing medical attention, patient self-management, education, and adherence to recommended medication (*Desta et al., 2021*). Statistics on diabetes-related medical problems are equally alarming (*American Diabetes Association, 2020*). In the Sub-Saharan region, the percentage of patients with diabetes sequelae varied, with retinopathy accounting for 7–63%, neuropathy for 27–66%, and
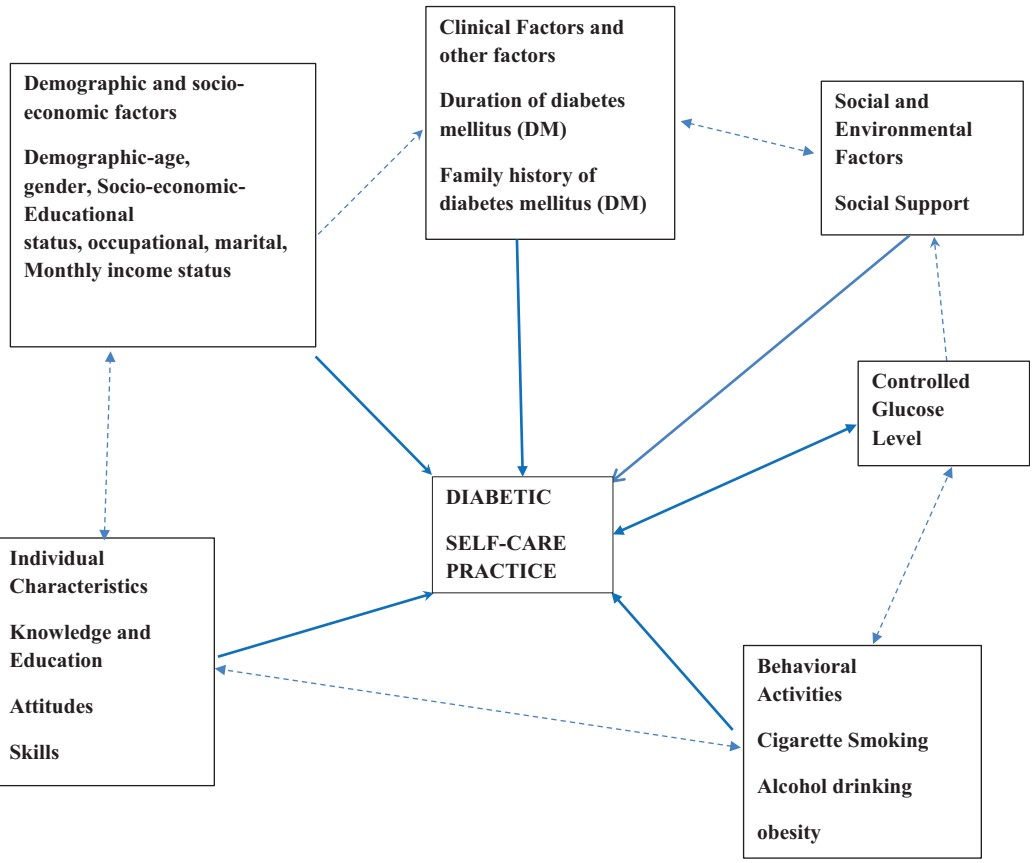

**Figure 1 Conceptual framework for the study to assess diabetes self-care practice and associated factors among type 2 diabetes patients attending public hospitals, in Bale zone, Oromia region, Ethiopia.**

nephropathy for 10–83% (*Heisler et al., 2005*). Diabetes may raise the risk of sepsis, pneumonia, and tuberculosis (*Heisler et al., 2005*).

One of the critical factors determining the evolution of diabetes and its associated consequences, which is primarily preventable, is patients' lack of knowledge of self-management tasks (*Heisler et al., 2005*). Since diabetes education enhances knowledge, attitudes, and skills that lead to better disease control, it is widely recognized as a crucial part of comprehensive diabetes self-management (*Świątoniowska et al., 2019*). Ineffective diabetes self-management continues to be a severe issue that affects communities and healthcare professionals everywhere. A patient's morbidity and mortality rate, the price of their prescription and laboratory tests, and the time and effort it takes to provide them with care all rise as a result of poor self-management. On the other hand, those who take good care of themselves live longer, get better results, experience fewer symptoms, and face positive outcomes (*Heisler et al., 2005*). Finding out how type 2 diabetes mellitus (T2DM) patients currently practice self-care is necessary, including medication adherence, dietary habits, physical activity, and compliance with blood glucose monitoring, to identify potential gaps and areas for improvement. Effective self-care habits, such as medication adherence, dietary adjustments, physical activity, and routine blood glucose monitoring,

are essential for the management of type 2 diabetes mellitus (*Ahmed et al., 2023*) as shown in the conceptual framework in Fig. 1. Although self-care practices are known to be crucial in managing diabetes, there is limited research on the extent and factors related to these practices at the zonal level. Accordingly, the purpose of this study is to evaluate the degree of self-care behaviours and the variables linked to them among patients with Type 2 diabetes mellitus (T2DM) who visit public hospitals in the Bale zone in the Oromia region of Ethiopia.

According to the research, there are differences in the knowledge and practices of people with type 2 diabetes in the field of individual and other treatment. To promote effective self-care for type 2 diabetes in the general population, health managers must be able to develop comprehensive and appropriate health promotion programs in the population. These studies will increase knowledge in this area and encourage others to assist in the self-care of people with diabetes. Additionally, the results of this research can form the basis for further research.

## METHODS AND MATERIALS

### Study area

Bale Zone is located in the Oromia Region in southeast Ethiopia, 430 kilometres from Addis Ababa. It comprises ten districts, covering a total area of 6,981.8 square kilometres, and is home to an estimated population of 1.4 million people. One general hospital, two district hospitals, a teaching hospital for referrals, 48 health centers, 201 health posts, and 26 private clinics make up the zone's healthcare system. The zone has been significantly affected by various crises, such as drought, epidemics, and flash floods.

During the Woreda health profile analysis, a total of 5,136 cases of diabetes mellitus were identified from the zonal hospitals' patient screenings, as reported in the Administration Zonal report (Fig. 2).

### Study design and period

A hospital-based cross-sectional study design was conducted in government hospitals in Bale Zone, Ethiopia, from February 05/2024–March 22/2024.

### Source and study population

#### Source population

The source population of the study included all type 2 diabetic patients who visit the diabetes mellitus clinics in the government hospitals located in Bale Zone.

#### Study population

The study population were all type 2 diabetic patients who were on treatment follow-up at the outpatient department during the study period from February 05/2024, up to March 22/2024.

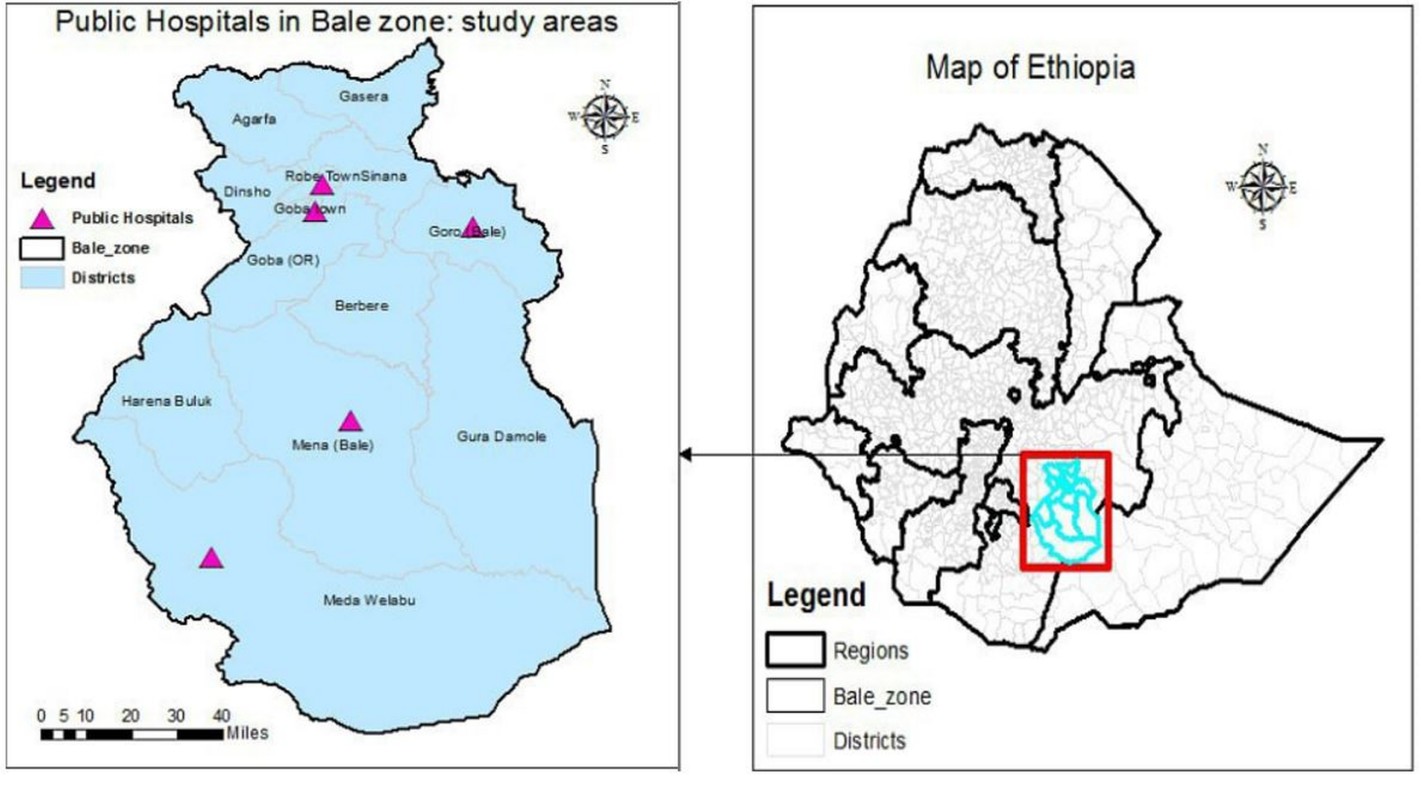

**Figure 2 Study setting area.**

## Inclusion and exclusion criteria

### Inclusion criteria

Every patient with type 2 diabetes who saw a physician during a data collection period between February 5, 2024, and March 22, 2024, and who had at least one follow-up before that date.

### Criteria for exclusion

✓ Patients in critical condition who are unable to give consent

✓ women with gestational diabetes

✓ Type 1 diabetes mellitus

✓ Children <18 years

## Determining the sample size

With a 95% confidence level, a 5% margin of error, and a 58% diabetes self-care practice (*Desalegn & Mukrima, 2016*) among diabetes patients, the sample size was determined using the single population proportion calculation.

**Number of samples (n)** $= \left\{ d^2 \right\}$

whereas: p = proportion of diabetes self care level or expected proportion = 0.58.
q = 1 − p; q = 1 − 0.58 = 0.42.

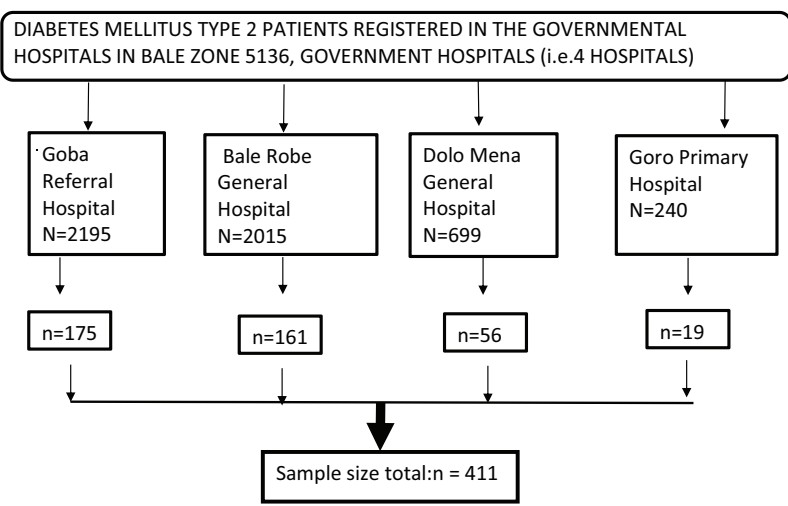

**Figure 3  Diagramatic study sample.**

n = the required sample size.

The margin of error = 5% (0.05) times d.

$n = (z.\alpha/2)^2 p (1 - p)/d^2$

$n = (1.96)^2 0.58 (1 - 0.58)/(0.05)^2 = 0.935/0.0025 = 374$. Accordingly, total sample size was determined to be = **411**.

## Sampling technique and sampling procedures

A systematic sampling strategy was used in the study to choose patients with type 2 diabetes, taking into account the patient flow in the month before data collection. The names of patients registered for hospital follow-up are listed in the diabetes mellitus ward registration book, which is where the patient list (sample frame) was sourced from. The sampling proportion in a month (N) was divided by the total number of samples (n) at the data collecting location (n′) to establish the sampling interval 'k', or k = N′/n′. Based on this interval, each Kth patient was then chosen for data collection. In the month before data collection, there were 1,287 cases of type 2 diabetes, and the sampling interval "k" was calculated as …1,287/411 ~ 3.

During the study period, there were four hospitals with a total of follow-up type 2 diabetes patients that were similar to the 5,136 cases from the previous year.

The patient list was taken from each follow-up clinic's registration records within the medical institutions. Using the following formula, samples were distributed proportionately to the total population of each stratum

Since N is the total population, ni = n/N * Ni.

Ni = total number of people in each stratum.

ni is the sample size for every stratum (Fig. 3).

## Methods for collecting data

### Data gathering tool

A systematic interviewer-administered questionnaire was used to gather the data. Before being distributed, the questionnaire was translated into Afan Oromo language, the native tongue. Socio-demographic information, clinical features of the respondents, diabetes knowledge, medication adherence, and diabetes self-care behaviours were all included in the questionnaire.

The Diabetes Self-Management Knowledge (DSCK-30) tool is a thirty-item measure designed to assess self-care knowledge in individuals with type 2 diabetes, utilizing yes/no response choices and revealing three scales of self-care proficiency through factor analysis (*Jackson et al., 2014*).

The Summary of Diabetes Self-Care Activities (SDSCA) measure is a brief self-report questionnaire on diabetes self-management that can be used as an additional self-care tool. It addresses many facets of diabetic self-management, including physical activity, blood glucose testing, good eating, foot care, and smoking (*Takele et al., 2021*).

### Supervisors and data collectors

A training session lasting one day was conducted at Robe General Hospital for supervisors and five data collectors. The principal investigator also provided ongoing awareness to ensure clarity during data collection.

A pre-test was carried out at Meda Welabu Primary Hospital to identify 20 diabetes patients and areas needing improvement. Necessary adjustments were made based on feedback received during the pre-testing to ensure clarity and accuracy.

Before patients could leave the facility, five BSc nurses proficient in Afan Oromo conducted face-to-face exit interviews to gather data. Data collectors cycled to different facilities during the collecting period to lessen bias. To protect patient anonymity, interviews were placed in a secluded area, apart from the main building. The lead investigator and two public health officers oversaw the data collection process.

## Operational definitions

**Self-care practice:** these are the daily tasks individuals undertake to manage their diabetes effectively. These tasks include monitoring blood glucose levels, following dietary recommendations, engaging in physical activity, taking prescribed medications, and seeking medical advice when necessary. The Summary of Diabetes Self-Care Activities (SDSCA) was expanded, and 18 questions including six items were utilized to measure self-care behavior (*Gul, 2010*). Those who scored ≥3 on the mean score were categorized as having a good level of self-care-behaviour by computing the overall level of self-care behaviour. Individuals who had an overall mean score of less than three were classified as practising poor self-care.

**Good *vs* poor diabetes self-management knowledge:** This refers to individuals' knowledge about managing diabetes. According to the DSCK-30 surveys, study participants who correctly answer 70% or more of the questions about self-care knowledge

are deemed to have a good grasp, while those who answer less than 70% are deemed to have a poor understanding of diabetes self-management (*Chinnappan et al., 2020*; *Hu et al., 2013*).

**Glycemic management in diabetes:** The level of glycemic control was indicated as 'good glycemic control' when fasting blood sugar (FBS) results were less than 130 mg/dl (7 mm/L) (*i.e.*, an average of four visits), 'poor glycemic control' takes place when a parameter is beyond the criteria of adequate glycemic control (*Jannasch, Kröger & Schulze, 2017*). It is ascertained from the patient's medical record in consultation with a physician in a chronic outpatient department during follow-up.

**Diet:** It is the type and quantity of food to be consumed each day, calorie intake, distribution of micronutrients, consumption of fruits and vegetables, and adherence to dietary recommendations for managing diabetes. After calculating the overall level of the self-care diet, scores above 1.4901 indicate high adherence to the recommended diet, and lower scores indicate poor adherence to the recommended diet (*Jannasch, Kröger & Schulze, 2017*).

**Physical activities:** the level and frequency of physical exercise individuals engage in, such as walking, jogging, cycling, or other forms of exercise. In this study, physical activities were measured in terms of duration, intensity, and consistency. Those who scored above the mean of 1.4954 had a good level of diabetes self-care practice (DSCP), and those who scored below the mean had a poor level of DSCP (*Sherifali et al., 2015*).

**Adherence to medication:** is the degree to which the patient takes the medication as directed. In this survey, a sum score of more than two denotes low medication adherence, a score between one and two denotes medium medication adherence, and 0 denotes great medication adherence (*Desalegn & Mukrima, 2016*).

**Social support:** refers to the presence and effectiveness of support systems that individuals have access to, such as family members, friends, healthcare professionals, or community resources, to maintain their diabetes self-care routine. A score above 1.15 indicates good social connectivity, while a score below 1.15 indicates poor social support (*Hasan, Ismail & Noor, 2024*).

**Diabetic comorbidity and complications present:** if a participant previously had diabetic complications like nephropathy, neuropathy, retinopathy, foot ulcers, heart disease, or other known issues, monitoring their health is crucial. It is ascertained from the patient's medical record in consultation with a physician in a chronic outpatient department during follow-up (*Scheurer et al., 2012*).

**Foot care:** regular maintenance of foot health practices customized to prevent complications from nerve damage and poor circulation. This includes foot examinations, daily care, proper footwear, and professional consultations. Individuals were asked about their daily foot care practices, such as using warm, soapy water, examining their feet for sores or redness, drying their feet thoroughly, and applying moisturizer. Those with a score

higher than the average of 1.1956 demonstrated good-foot-care practices (*Hirpha, Tatiparthi & Mulugeta, 2020*).

**Current tobacco smokers:** the habit of tobacco smoking—measured in terms of frequency (number of cigarettes per day), duration of smoking history, attempts at quitting, and awareness of the detrimental effects of smoking on diabetes control—was assessed among those who used tobacco products within the previous year (*Dereje et al., 2020*).

## Quality control of data

To ensure the quality of translation to Afan Oromo language, the questionnaire was pretested on twenty type-2 diabetic patients at a follow-up visit at Meda Welabu district hospital before data collection. The required adjustments were made, and supervisors were paired with particular patients to track down and fix any issues with the quality of the data.

Following the gathering of data, an initial frequency analysis was conducted to identify any missing variables within the data set. This step is crucial as it allows researchers to recognize gaps in the data that could affect the overall analysis and conclusions drawn from the study. Once any missing variables were identified, the raw data were adjusted accordingly to ensure completeness and accuracy.

## Data analyzing

The KoBotool box was used to gather the data, which was then exported to SPSS version 26. Tables, frequencies, and graphs were used to summarize the descriptive statistics that were calculated for each variable. Utilizing binary logistic regression, variables related to the result variable were found. It was calculated to get the crude odds ratio (COR) and 95% confidence interval (CI). The multivariable logistic regression analysis was then applied to the variables whose $P$-value was less than 0.25. In the context of multivariable logistic regression models, a $P$-value of less than 0.05 was deemed statistically significant. A 95% confidence interval was computed for the adjusted odds ratio (AOR).

Data cleaning and transformation were performed for medication adherence (mean 1.6624), diet (mean 1.4901), physical activities (mean 1.4954), SMBG (mean 1.4964), foot care (mean 1.4672), diabetes self-management knowledge (1.1598), and smoking (1.2701) variables using the mean value as cut off. High values were recorded as (1) indicating a good level of DSCP, and lower mean scores were recorded as (2) representing a poor level of DSCP.

## Ethical consideration

Protocol number SPH/296/2024 was used to get ethical approval for the study from the Addis Ababa University School of Public Health IRB. Additionally, letters of support were sent to ORHB and the Oromia Health Bureau for each of the public hospitals in the Bale Zone. Furthermore, the Oromia Public Health Research and Emergency Management Directorate, reference number BFO/116/1314, provided ethical clearance. After informing the respondents about the aim and purpose of the study, verbal consent was obtained before delivering the questionnaire. Each participant gave their informed consent after being made aware of their right to withdraw from the activity and their freedom to stop at

any moment. To maintain confidentiality and privacy, the study subject's name was not included in the questionnaire.

## RESULTS

### Socio-demographic characteristics

The study achieved a 100% response rate from the 411 distributed questionnaires. Among patients with diabetes, the most prevalent age groups were 50–59 years (24.1%, $n = 99$), 60–69 years (20.7%, $n = 85$), 40–49 years (20.4%, $n = 84$), and over 70 years (19%, $n = 78$). The patient's average age was 54.28 (SD ± 15.4). The participants were between the ages of 18 and 108. The educational status of the participants followed this pattern: 129 (31.4%) had attended primary education, 97 (26.5%) were not able to read and write, 66 (16.1%) had participated in secondary education, and 53 (20.6%) had attended grade 12 or higher. Regarding marital status, 341 (83%) respondents were currently married, while 27 (6.6%), 25 (6.1%), and 18 (4.4%) were widowed, single, and divorced, respectively. Out of the total respondents, 209 (50.9%) were unemployed. The results on the level of monthly income of the respondents revealed that 289 (70.3%) earned more than 1,000 birr, followed by 65 (15.8%) who earned between 501 and 1,000 birr. The study participants' average monthly income was 5,411.84 (SD ± 12,037.455).

### Clinical and other factors related to diabetes

The duration of disease among respondents varied, with the largest group (45.7%, $n = 188$) experiencing it for less than 5 years, while 35% ($n = 144$) had a duration between 5 and 10 years. The mean duration of diabetes was 7.34 years (SD ± 15.4), ranging from a minimum of 1 year to a maximum of 35 years. Doctor-verified long-term diabetes problems were reported by 53.5% of the total respondents ($n = 220$). A majority of the 356 respondents had received diabetic self-care education at least once. Of those who received this instruction, the majority (82%) identified their healthcare providers as the source, followed by the media (11.5%). The finding that 84.9% of participants do not belong to the local chapter of the diabetes association underscores significant areas for improvement in community engagement and resource accessibility related to diabetes management.

### Type 2 diabetes mellitus patients' overall degree of diabetes self-care practice (DSCP)

#### Diabetes self-management knowledge

Among the study participants, 158 (38.4%) demonstrated a high level of overall knowledge, while the remaining 253 (61.6%) exhibited a lower understanding of self-care practices. The percentages of correct responses regarding self-care knowledge were 38.2% for modifiable lifestyles, 61.3% for adherence, and 29.7% for consequences of uncontrolled blood glucose levels. A majority, 353 (85.9%), acknowledged that the fasting blood sugar (FBS) test is suitable for monitoring blood sugar control over a period of 2–3 months. Furthermore, 89.9% of participants agreed that only qualified medical professionals in

**Table 1** Overall performance of respondents in evaluating the level of medication adherence, knowledge, and self-care advice among patients with type 2 diabetes mellitus attending government hospitals in the Bale zone, Oromia region, Ethiopia, in 2024 ($n = 411$).

| Characteristics | Yes | No |
|---|---|---|
| Reduced thirst, less frequent urination, and higher energy are common in those whose blood sugar levels are near normal. | 267 (65%) | 144 (35%) |
| Extended hyperglycemia can cause visual impairments or possibly blindness. | 335 (81.5%) | 76 (18.5%) |
| Extended periods of uncontrolled blood sugar can lead to heart attacks, strokes, and kidney problems. | 333 (81%) | 78 (19%) |
| Signs of elevated blood sugar include shaking, disorientation, altered behaviour, and perspiration. | 288 (70.1%) | 123 (29.9%) |
| Before and after any physical activity, one should check their blood glucose level. | 369 (89.8%) | 42 (10.2%) |
| Blood sugar control during a 2 to 3-month period can be monitored with a fasting blood sugar (FBS) test. | 353 (85.9%) | 58 (14.1%) |
| A diabetic needs to perform daily oral hygiene measures, such as brushing and flossing. | 369 (89.8%) | 42 (10.2%) |
| One with diabetes can benefit from wearing a tight elastic hose or socks. | 342 (83.2%) | 69 (16.8%) |
| A diabetic can monitor and respond to changes in their blood sugar levels with the use of self-blood glucose monitoring (SBGM). | 334 (81.3%) | 77 (18.7%) |
| Making plans for a diabetic patient to reach their objectives should only be done by doctors. | 362 (88.1%) | 49 (11.9%) |
| It is not crucial to maintain a healthy weight when managing diabetes. | 354 (86.1%) | 57 (13.9%) |
| Physicians and other medical professionals can collect information for treatment planning with the use of self-blood glucose monitoring (SBGM). | 360 (89.8%) | 51 (12.4%) |
| Only a licensed medical practitioner and other hospital health professionals should check the blood pressure and blood sugar of a diabetic patient. | 369 (89.8%) | 42 (10.2%) |
| It is necessary to engage in physical activity for 20-30 min a day, a minimum of 3 days a week. (A few examples of physical activities are brisk walking, housework, and stair climbing.) | 375 (91.2%) | 36 (8.8%) |
| Any changes in a diabetic patient's vision should be reported to their physician. | 283 (68.9%) | 128 (31.1%) |
| When a diabetic is unable to alter a certain lifestyle, they and their physician should come to a mutually agreed upon decision. | 378 (92%) | 33 (8%) |
| Particularly when trimming their toenails, a diabetic should take additional care of their feet. | 281 (68.4%) | 130 (31.6%) |
| Frequent exercise doesn't make insulin or other diabetic medications less necessary. | 336 (81.8%) | 75 (18.2%) |
| A diabetic patient should only seek assistance from their medical team when they become ill. | 351 (85.4%) | 60 (14.6%) |
| When starting insulin therapy for a diabetic who may need it, the patient should receive the proper guidance on meals and Self Blood Glucose Monitoring (SBGM). | 338 (82.2%) | 73 (17.8%) |
| Cigarette smoking can aggravate diabetes | 344 (83.7%) | 67 (16.3%) |
| In a diabetic, blood glucose monitoring is more crucial than blood pressure monitoring. | 317 (77.1%) | 94 (22.9%) |
| Do you remember to take your diabetes medication regularly? | 152 (37%) | 259 (63%) |
| Other than simple forgetfulness, people occasionally fail to take their prescriptions as prescribed. During the last 2 weeks, can you recall any days when you neglected to take your medication? | 92 (22.4%) | 319 (77.6%) |
| Have you ever reduced or discontinued taking your medication without consulting your physician because you had worsening symptoms while using it? | 56 (13.6%) | 355 (86.4%) |
| Are you sometimes, when you travel or leave home, in a hurry and forget to take your diabetes medication? | 129 (31.4%) | 282 (68.6%) |
| Have you taken all of your medication yesterday? | 348, or 84.7% | 63 (15.3%) |
| Do you ever stop taking your medication when you think your symptoms are under control? | 73 (17.8%) | 338 (82.2%) |
| For some people, taking medication daily can be a real inconvenience. Do you sometimes find it difficult to follow your treatment plan? | 135—or 32.8% | 276, or 67.2% |
| How often does it get hard for you to remember to take all of your medications? | 125 (30.4%) | 286 (69.6%) |

(Continued)

| Table 1 (continued) | | | |
|---|---|---|---|
| **Characteristics** | | **Yes** | **No** |
| 1  Diet management | Maintain a low-fat diet | 310 (75.4%) | 101 (24.6%) |
| | Follow a complex carbohydrate diet | 218 (53%) | 193 (47%) |
| | Cut back on your calorie intake to help you lose weight | 234 (56.9%) | 177 (43.1%) |
| | Consume a lot of foods high in dietary fibre. | 314 (76.4%) | 97 (23.6%) |
| | Consume a large amount of fruits and vegetables—at least five servings each day. | 154 (37.5%) | 257 (62.5%) |
| | Consume very little sugar-filled foods (such as desserts, regular soda, and candy bars) | 158 (38.4%) | 253 (61.6%) |
| | My healthcare team has not given me any advice about my diet | 79 (19.2%) | 332 (80.8%) |
| 2  Physical exercise | Engage in light activity daily basis, such as walking | 337 (82%) | 74 (18%) |
| | At least 3 times a week, engage in continuous exercise for at least 20 min. | 193 (47%) | 218 (53%) |
| | Include exercise in your everyday schedule by, for instance, parking a block away and walking, or using the stairs rather than the elevator | 275 (66.9%) | 136 (33.1%) |
| | Exercise to a set quantity, kind, length, and intensity | 157 (38.2%) | 254 (61.8%) |
| | I have not been given any recommendations about exercising by my healthcare team | 75 (18.2%) | 33 (68.8%) |
| 3  Sugar level | Using a colour chart and a drop of blood from your finger, check your blood sugar | 287 (69.8%) | 124 (30.2%) |
| | Use the machine to measure your blood sugar and interpret the findings | 188 (45.7%) | 223 (54.2%) |
| | Test your urine for sugar | 259 (63%) | 152 (37%) |
| | I have not received any guidance regarding my blood or urine sugar level from my medical staff. | 94 (22.9%) | 317 (77.1%) |
| 4  Smoking | Did anyone inquire about your smoking status during your most recent medical visit? | 173 (42.1%) | 238 (57.9%) |
| | If you smoke, did someone advise you to give up at your most recent appointment with a doctor? | 145 (35.3%) | 266 (64.7%) |
| | Do not smoke (Never Smoke) | 309 (75.2%) | 102 (24.8%) |
| | When was the last time you had a cigarette? never used tobacco products | 209 (50.9%) | 202 (49.1%) |
| | When was the last time you had a cigarette? Over 2 years prior | 21 (5.1%) | 390 (94.9%) |
| | When was the last time you smoked a cigarette? 2 to 3 years prior | 14 (3.4%) | 397 (96.6%) |
| | When was the last time you had a cigarette? Between 4 and 12 months ago | 15 (3.6%) | 396 (96.4%) |
| | When was the last time you had a cigarette? Between 1 and 3 months prior | 14 (3.4%) | 397 (96.6%) |
| | When was the last time you had a cigarette? In the previous month | 12 (2.9%) | 399 (97.1%) |
| | When was the last time you had a cigarette? Now | 11 (2.7%) | 400 (97.3%) |

hospitals should be responsible for checking a diabetic patient's blood sugar and blood pressure. Notably, 81.6% of participants believed that regular exercise can decrease the reliance on insulin or other diabetic medications.

### Medication adherence

Adhering to medication guidelines is crucial for effective diabetes management as it significantly influences treatment outcomes. This aspect evaluates how well individuals comply with their prescribed medication regimen, including dosage schedules, intake frequency, and adherence to medical instructions. Poor medication adherence may result in uncontrolled blood sugar levels and an elevated risk of complications.
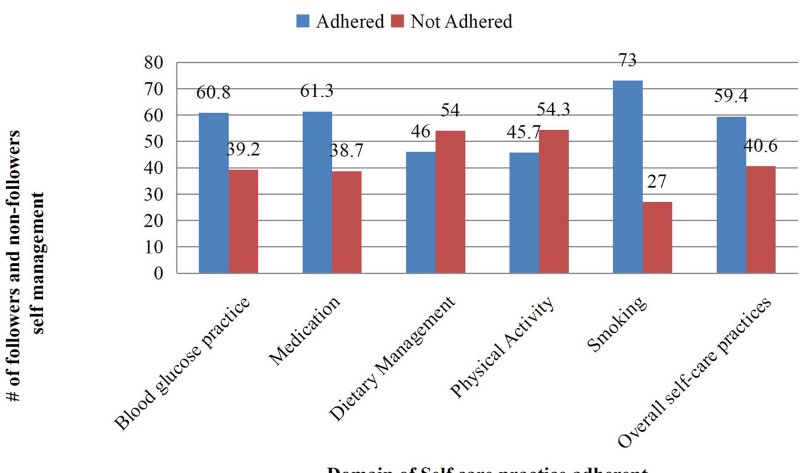

**Figure 4 Domain of self-care practice.**     

### Diabetes self-care practice

A majority of respondents (60.8%) demonstrated proper self-monitoring of blood glucose practices among the 250 surveyed. A total of 252 (61.3%) respondents adhered to anti-diabetic medication guidelines, while more than half, 222 (54%), did not comply with recommended dietary management practices. A total of 188 people (45.7%) said they participated in physical activities that complied with recommendations. The majority (73%) followed advice for smoking cessation. The overall self-care practices (SDSCA) rating was 244 (59.4%). Additionally, 81% of respondents said they had never received advice on managing their nutrition, 81.8% said they had never received direction on exercising, and 77.1% said they had never received expert advice or instruction on monitoring their blood or urine glucose levels as shown in Table 1 and Fig. 4.

## FACTORS RELATED TO OVERALL SELF-CARE ACTIVITIES (SDSCA)

In this study, 40.6% of individuals ($n$ = 167) did not practice self-care, while 59.4% ($n$ = 244) reported engaging in self-care activities. Bivariate analysis, utilizing a significance threshold of $P \leq 0.2$, identified several variables for inclusion in a multivariable binary regression model: sex, length of diabetes, education and employment status, diabetic health education, membership in a diabetic association, monthly income, and understanding of self-care. Further analysis revealed significant associations between the extent of self-care routines (SDSCA) and monthly income, occupational status, diabetic complications, diabetic health education, membership in a diabetic association, and self-care knowledge. Specifically, individuals with a monthly income exceeding 1,001 birr demonstrated a 2.39 times greater likelihood of maintaining self-care routines compared to those earning less than 500 birr (adjusted odds ratio (AOR) = 2.39, 95% confidence interval (CI) [1.19–4.80], $P < 0.014$).

**Table 2 Domains of self-care practice.**

| The domain of self-care practices | Good percentage (95% CI) | Low percentage (95% CI) |
|---|---|---|
| Diet | 46 [39.4–52.5] | 54 [46.8–61.1] |
| Exercise | 45.7 [39.1–52.2] | 54.3 [47.1–61.4] |
| Sugar level | 60.8 [53.2–68.3] | 39.2 [33.1–45.2] |
| Smoke | 73 [64.7–81.2] | 27 [21.9–32] |
| Medication | 59.4 [51.5–66.4] | 40.6 [34.4–46.7] |
| Foot care | Good 53.3% [46.2–60.3] | Poor 46.7% [40.0–53.3] |
| Overall self–care practice | 61.4 [57.5–65.2] | 38.79 [32.6–44.7] |

**Table 3 Factors related to self-care practice.**

| Variable | Categories | Level of SDSCA | | The crude OR (95% CI) | The AOR (95% CI) |
|---|---|---|---|---|---|
| | | Practicing good self-care | Ineffective self-care habits | | |
| Gender | Men | 139 | 106 | 1.31 [0.82–1.96] | 1.26 [0.76–2.07] |
| | Women | 105 | 61 | 1 | 1 |
| Educational Status | Illiterate | 58 | 39 | 1 | 1 |
| | Only able to read and write | 27 | 39 | 1.30 [0.65–2.62] | 1.75 [0.72–4.23] |
| | Primary | 81 | 48 | 2.80 [1.32–5.95]* | 2.24 [0.94–5.36] |
| | Secondary | 43 | 23 | 1.15 [1.00–3.56]* | 1.24 [0.58–2.66] |
| | Above grade 12 | 35 | 18 | 1.04 [0.48–2.22] | 0.99 [0.43–2.27] |
| Monthly income | <500 Birr | 49 | 16 | 1 | 1 |
| | 501–1,000 Birr | 44 | 13 | 0.90 [0.39–2.09] | 0.73 [0.29–1.85] |
| | >1,001 Birr | 151 | 138 | 2.79 [1.52–5.15]* | 2.39 [1.19–4.80]* |
| Occupation | Government | 87 | 46 | 1.34 [0.85–2.11] | 1.25 [0.71–2.19] |
| | Private | 122 | 87 | 1.83 [1.01–3.32]* | 2.09 [1.06–4.13]* |
| | *Others | 35 | 34 | 1 | 1 |
| Being a member of the diabetic association | Yes | 45 | 17 | 1.99 [1.09–3.62]* | 1.85 [0.93–3.67]* |
| | No | 199 | 150 | 1 | 1 |
| Duration of diabetes | <5 | 129 | 59 | 1 | 1 |
| | 5–10 | 65 | 79 | 0.78 [0.45–1.36] | 0.96 [0.52–1.79] |
| | >10 | 50 | 29 | 2.09 [1.19–3.68]* | 1.82 [0.97–3.42]* |
| Diabetic health education about diabetes self-care | Yes | 200 | 156 | 3.12 [1.56–6.24]* | 2.85 [1.33–6.12]** |
| | No | 44 | 11 | 1 | 1 |
| Self-care knowledge | Yes | 78 | 80 | 1.95 [1.30–2.93]* | 2.04 [1.24–3.34]* |
| | No | 166 | 87 | 1 | 1 |
| Diabetic-related long-term complications | Yes | 112 | 108 | 1 | 1 |
| | No | 132 | 59 | 2.15 [1.43–3.23]** | 2.68 [1.64–4.36]** |

Notes:
*statistically significant at 0.05 level of significance.
**statistically significant at 0.01 level of significance.

Compared to people in other occupations, private employees had a 2.09-fold higher likelihood of practising self-care (AOR, 95% CI, 2.09, [1.06–4.13], $P < 0.033$). Adherence to self-care practices was consistently linked to receiving diabetes health education. The likelihood of adhering to self-care behaviours was 2.85 times higher for those who received such education (AOR, 95% CI, 2.85, [1.33–6.12], $P < 0.007$). Individuals who belonged to the diabetic association had a 1.85-fold higher likelihood of following through on self-care routines compared to those who did not (AOR = 1.85, [0.93–3.67], $P < 0.077$). Self-care practices were adhered to 2.04 times more frequently by individuals with excellent knowledge of self-care than by those with low knowledge (AOR, 95% CI, 2.04, [1.24–3.34], $P < 0.004$). Self-care behaviours were 2.68 times more common among participants without diabetes problems (AOR = 2.68, 95% CI [1.64–4.36], $P < 0.000$). The distribution of self-care behaviours among patients in public hospitals in the Bale zone of the Oromia region, Ethiopia, is shown in Table 2.

It covers aspects like diet, exercise, sugar levels, smoking, medication usage, and foot care, with percentages for favourable and unfavourable practices and their corresponding 95% confidence intervals. This data aims to offer a comprehensive understanding of self-care practices in this population across various domains Table 3.

✓ To adjust the model for AOR, we used techniques such as variable selection, transformation, interaction terms, and handling missing data.

✓ The significant candidate variables for the multivariable model at a 0.25 level are: monthly income, education, occupational status, diabetic health education, membership in the diabetic association, sex, diabetic complications, duration of diabetes, diabetes self-management knowledge.

## DISCUSSION

The self-care routines of individuals with type 2 diabetes mellitus are not well documented in Ethiopia. As a result, the goal of this study was to evaluate type 2 diabetes patients receiving care at government hospitals in Ethiopia's Bale Zone, Oromia Region for their level of self-management and related characteristics. In this survey, the age range of 50 to 59 years was found to include the majority of participants (24.1%), while the age range of 60 to 69 years comprised 20.7% of the respondents. Likewise, research conducted in Egypt revealed 66% and 44%, respectively (*Tan & Magarey, 2008*).

Of the 250 responders in this survey, the majority (60.8%) followed the recommended guidelines for self-monitoring blood glucose. This result is higher than earlier research in Ethiopia (5%), India (3%), Malaysia (15%), and Nigeria (8%) but lower than the U.S.A. (78%) rate (*Tan & Magarey, 2008*; *Eregie & Unadike, 2011*; *Beckles et al., 1998*; *Feleke & Enquselassie, 2006*; *Moussa et al., 2023*).

A Malaysian study found that SMBG practice was significantly predicted by education level, monthly income, long-term issues connected to diabetes, and marital status (*Tan & Magarey, 2008*). Similarly, this study found a significant correlation between SMBG practices and education level, marital status, monthly income, and long-term issues associated with diabetes. Even though self-monitoring of blood glucose, or SMBG, is

acknowledged as helpful and efficient in achieving diabetes management, this study discovered that only a small percentage of respondents with diabetes were engaging in SMBG activities. Financial obstacles preventing the acquisition of the gadget and its strips and a lack of knowledge about its significance in diabetes treatment are probably the causes of this lack of training (*Tan & Magarey, 2008*).

A total of 38.7% of participants in this study were unable to take their prescribed medication as directed, which was less than the results of studies conducted in Malaysia (46%) and Nigeria (46%) (*Tan & Magarey, 2008*; *Eregie & Unadike, 2011*). The study found a strong correlation between monthly income, adequate self-care knowledge, long-term issues connected to diabetes, and medication adherence. However, a Nigerian study found that non-adherence was substantially correlated with elements including pharmacological side effects, perceived inefficacy of prescribed drugs, and lack of financial means (*Tan & Magarey, 2008*; *Eregie & Unadike, 2011*). According to the study, more individuals followed their medication regimen than there were non-adherents, which may have been caused by their knowledgeable attitude and favourable opinion of prescribed medications, especially insulin injections (*Tan & Magarey, 2008*; *Eregie & Unadike, 2011*).

Merely 46% of the subjects in the study adhered to the suggested dietary guidelines (*Mahfouz & Awadalla, 2011*). This is less than the rates in other nations, such as Iran (96% for males and 100% for women), Egypt (81%), and India (52% for women and 32% for men). According to the study, there could be several reasons for the lower adherence rate, including financial constraints, a skewed understanding of the value of fruits and vegetables, a lack of knowledge about healthy eating programs, and cultural and lifestyle disparities. A large number of respondents also said they were unable to create or adhere to a healthy food plan (*Moussa et al., 2023*; *Mahfouz & Awadalla, 2011*; *Yekta et al., 2011*; *Al-Kaabi et al., 2008*).

A study conducted in Egypt found a statistically significant correlation between married status and adherence to dietary management of diabetes; of those who were married, nearly a quarter (26%) adhered to dietary management. The study also found that longer disease duration and the 40–49 age group had a positive effect on adherence to dietary management practices (*Mahfouz & Awadalla, 2011*). Similar to the previous study, this one also revealed that there was no significant correlation between people without diabetic problems and high levels of education, which may have been because of the small sample size.

Compared to studies conducted in the United Arab Emirates, where only 3% of males and 4% of females adhered to the recommended guidelines, 45.7% of respondents in this study followed the recommended guidelines for physical activity. There is a possibility that this difference in results stems from the fact that most patients do not lead sedentary lives and instead engage in physical activity daily, even if it is just taking a short walk for 30 min a day (*Al-Kaabi et al., 2008*; *Mahamar et al., 2021*). That is comparable, though, to research from Malaysia (46%), and the United States (52%), where 52% of participants reported exercising once a week or more (*Moussa et al., 2023*; *Al-Kaabi et al., 2008*).

A study conducted in Malaysia found a substantial correlation between age, self-care behaviours, anti-hyper-glycemic medication type, and education level (*Tan & Magarey,*

*2008*). Nonetheless, this research demonstrated a strong correlation between diabetes health and monthly income, educational attainment, and employment position, instruction on diabetic self-management (*Tan & Magarey, 2008*).

A total of 59.4% of participants in this research have good overall self-care routines across all dimensions. Compared to studies conducted in Finland (81%), and Iran (74%), where participants adhered to the overall self-care practice dimensions, this outcome is lower. This study's finding is lower than that of the other research; possible explanations include financial constraints, a lack of understanding of the significance of the activities, societal variance, and lifestyle disparities (*Yekta et al., 2011*; *Toljamo & Hentinen, 2001*). An Iranian study found that the length of diabetes, monthly income, and diabetic complications all positively correlated with the degree of self-care practice (*Yekta et al., 2011*; *Toljamo & Hentinen, 2001*). Additionally, it was discovered that participation in the diabetic association, monthly income, employment status, diabetic problems, and diabetic health education regarding diabetes self-care is crucial for enhancing the practice of self-care. According to a Finland study, adherence to self-care behaviours was not substantially correlated with diabetic complications (*Yekta et al., 2011*; *Toljamo & Hentinen, 2001*).

### Limitations

The cross-sectional design limits clarity of causal relationships and the narrow focus on public hospitals restricts generalizability. Recall bias and socially desirable responses compromise data accuracy. Additionally, uncertainty in causality between behaviors and factors, inadequate representation of diverse groups, and lack of extended follow-up periods are potential limitations.

## CONCLUSION

Bale, Ethiopia T2D patients exhibit inadequate diabetes self-care habits. Significant links exist between self-care behaviors and factors like socioeconomic status, access to education, affiliation with diabetic associations, and self-care knowledge. Those with lower incomes and non-private jobs need tailored support. Encouraging diabetic association membership and enhancing education access are vital for better self-care. Improving knowledge and preventative care for complications is essential for diabetes management in Bale Zone public hospitals.

### Recommendations

Strategies must be coordinated and monitored to improve outcomes for T2DM patients in Bale Zone hospitals.

– Increase access to health education through a multidisciplinary approach and the information, education and communication (IEC) program. Collaborate with healthcare professionals, public health authorities, academic institutions, and community health workers.

– Raise awareness of self-care practices and promote them using the information, education and communication (IEC). Involve healthcare providers, advocacy organizations, pharmaceutical companies, and government health agencies.

– Focus on diabetic education for patients with limited financial resources. Engage non-profit organizations, government health bodies, community health centers, and social workers.

– Encourage participation in diabetic associations to enhance self-care knowledge. Collaborate with diabetes associations, specialized healthcare providers, and patient support networks.

– Ensure timely detection and management of complications to support adherence to self-care. Collaborate with general practitioners, endocrinologists, diagnostic facilities, and insurers for preventive screenings.

– Future studies should use longitudinal designs to explore the relationship between self-care practices and influencing factors.

– Address social desirability bias and ensure confidentiality during data collection.

– Healthcare providers can utilize these findings to develop targeted interventions for T2DM patients in Bale Zone.

## ACKNOWLEDGEMENTS

We want to express our profound gratitude to the academic staffs, Mr. Abdulnasir Abagero who have supported the filed work. We also want to thank the field supervisor Gadissa Mulatu and Kibatu Merhaba.

### Funding

The authors received no funding for this work.

### Competing Interests

The authors declare that they have no competing interests.

### Author Contributions

- Eshetu Zemen conceived and designed the experiments, performed the experiments, analyzed the data, prepared figures and/or tables, authored or reviewed drafts of the article, and approved the final draft.
- Yimer Seid Yimer conceived and designed the experiments, performed the experiments, analyzed the data, prepared figures and/or tables, authored or reviewed drafts of the article, have more closely guided author than others, and approved the final draft.
- Negussie Deyessa Kabeta conceived and designed the experiments, performed the experiments, analyzed the data, prepared figures and/or tables, authored or reviewed drafts of the article, and approved the final draft.
- Yonas Abebe conceived and designed the experiments, performed the experiments, analyzed the data, prepared figures and/or tables, authored or reviewed drafts of the article, and approved the final draft.

## Human Ethics

The following information was supplied relating to ethical approvals (*i.e.*, approving body and any reference numbers):

The School of Public Health, Addis Ababa University granted Ethical Approval to carry out the study.

## Ethics

The following information was supplied relating to ethical approvals (*i.e.*, approving body and any reference numbers):

Oromia regional health and research directorate.

## Data Availability

The raw data is available in the Supplemental Files.

## Supplemental Information

Supplemental information for this article can be found online at http://dx.doi.org/10.7717/peerj.19529#supplemental-information.

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
