# Peer review of "Self-care practices and associated factors among type 2 diabetes mellitus patients attending public hospitals in Bale zone, Oromia region, Ethiopia"

_PeerJ, doi:10.7717/peerj.19529_

## Round 0.1 · original submission · Minor Revisions

Dear Authors,

Thank you for submitting your manuscript titled "Self-Care Practices and Associated Factors Among Type 2 Diabetes Mellitus Patients Attending Public Hospitals in Bale Zone, Oromia Region, Ethiopia" to PeerJ. After thorough peer review, we acknowledge the originality and relevance of your study. However, the reviewers have identified areas that require improvement before publication.

Based on their evaluations, my decision is Minor Revisions. Please refine the language to align with professional and academic standards. Ensure that the research question is clearly stated and that the methodology is more thoroughly justified, particularly in terms of study design, sample size determination, and data collection procedures. Ethical clearance and validation of instruments should also be explicitly mentioned. Additionally, further discussion is needed on how identified factors, such as socioeconomic status and access to diabetes education, influence self-care behaviors. The limitations of the study, particularly those related to its cross-sectional design, should be acknowledged.

Reviewers have also suggested structural and grammatical corrections throughout the manuscript, including improving paragraph organization, refining subheadings for consistency, and ensuring clarity in data presentation. Additionally, any abbreviations should be properly introduced before use, and inconsistencies in terminology should be corrected.

Please revise the manuscript accordingly and submit a detailed response addressing the reviewers' comments. We look forward to receiving your revised submission within the designated timeframe.

Best regards,

Dr. Gustavo Pedrino
Editor, PeerJ

Reviewer 1 ·

Basic reporting

The authors used unambiguous English but need to use acceptable professional and research languages.
Literature references are adequate. The background is sufficient.

Experimental design

The study is original.
The research question should be well stated.
The methodology should be improved upon. A convincing justification for the choice of the design should be included. Adequate and clear information on sample size selection should be given.

Validity of the findings

The findings are clear. The study can be replicated.

Additional comments

Reviewer’s Comments on Self-Care Practices and Associated Factors Among Type 2 Diabetes Mellitus Patients Attending Public Hospitals in Bale Zone, Oromia Region, Ethiopia. (#107806)
BASIC REPORTING
ABSTRACT
Background: Put the components of self-care in brackets since you have many of them (lines 14 & 15).
Objective: should be stated in past tense because the study has been concluded (line 16).
Methods: The number of public hospitals and the justification for choosing them should be given. Ethical clearance and validity of the instruments should be mentioned too (lines 18 & 19).
Conclusion: Diabetic education should be diabetes (lines 30 & 31).
Keywords: Use title cases for the key words and arrange them alphabetically (lines 37).
INTRODUCTION
…... diabetes can be caused by many factors; urbanization (put semi-column after factors instead of full-stop, line 46).
Put – in between 70 & 120 not & self-monitoring _ (lines 49 & 51).
METHODOLOGY
More information should be given on justification for the choice of research design, number of hospitals and health posts used (this is important because specialized DM care can’t be given in all the health posts), this is also important given the time used for data collection. The name of the formula used for sample size calculation should be written. Why did you not use attrition rate?
Sampling Technique and Sampling Procedures
Change are to were (Line 46)
Reframe lines 152 & 153.
Change “before being distributed” on lines 163 & 164 to “before distribution”.
… the questionnaire was translated into Afan Oromo (add language, line 164).
Re-cast the sentence on lines 167 & 168.
Re-cast the sentences on lines 181-185. Don’t start a sentence with a preposition like you have on line 183.
Lines 187-196 are not necessary, delete them.
Lines 197-248 are not needed. If you want to define operational terms for the study, define only the keywords and let them come immediately after introduction.
Merge lines 254-258 (an ideal paragraph is between 5 & 8 lines) and correct the grammatical errors in the last paragraph.
Correct your sentence on line 265.
Correct your sentences that began with prepositions under ethical considerations (lines 274-282).
Delete everything from 283-286. They are not needed.
RESULTS, DISCUSSION AND OTHERS
You can’t begin a paragraph or a sentence with figures. You can write in words or use something like Firstly. Use mean age instead of age range.
Use title cases for all sub-headings consistently. e.g line 303.
Use – appropriately instead of _
Correct all your grammatical errors
Merge too short paragraphs, (an ideal paragraph is between 5 and 8 lines).
Change diabetic to diabetes where necessary.
Change Limitation to Limitations because you have more than one.
Write the full meaning of IEC before abbreviating under recommendations.
Change recommendation to recommendations because you have more than one.
Correct all grammatical errors up to the acknowledgements. Worthy of note is: members of academic staff which should replace academic staffs.

Reviewer 2 ·

Basic reporting

The study is well-structured, with a clear flow from the background to the methodology, results, and conclusion.

Experimental design

The study employs a well-thought-out experimental design.

Validity of the findings

The validity of the findings is strengthened by the use of a large, representative sample of 411 type 2 diabetes patients, which helps ensure that the results are reflective of the broader population within public hospitals in Bale Zone.

Additional comments

The manuscript "Self-Care Practices and Associated Factors Among Type 2 Diabetes Mellitus Patients Attending Public Hospitals in Bale Zone, Oromia Region, Ethiopia" by Zemen et al. highlights that type 2 diabetes patients in Bale, Ethiopia, exhibit inadequate diabetes self-care practices, with significant associations between self-care behaviors and factors such as socioeconomic status, access to diabetes education, membership in diabetic associations, and overall self-care knowledge. This study has the potential to guide future public health initiatives aimed at improving diabetes management in underserved regions.
I recommend publication of this paper with a few minor issues being addressed.
Point 1: there is a lack of depth in explaining how the identified factors (such as income and employment status) specifically impact self-care behaviors.
Point 2: There is also no mention of potential limitations in the study, such as the cross-sectional nature of the design, which prevents any causal inferences from being made.
Point 3: The methodology needs more detailed explanation to ensure the study’s findings are valid and generalizable. The study risks offering recommendations that may not be feasible or applicable in the real-world context of Bale Zone.

---

## Round 0.2 · accepted · Accept

Dear Eshetu Zemen,

Thank you for submitting the revised version of your manuscript entitled “Self-Care Practices and Associated Factors Among Type 2 Diabetes Mellitus Patients Attending Public Hospitals in Bale Zone, Oromia Region, Ethiopia” (Manuscript ID #107806) to PeerJ.

I have carefully reviewed the detailed point-by-point responses to the reviewers’ comments, as well as the revised manuscript. I appreciate the effort made to address the feedback constructively and comprehensively. The revisions have significantly improved the clarity, rigor, and overall quality of the study.

In light of the satisfactory responses and the improved manuscript, I am pleased to inform you that your article has been accepted for publication in PeerJ.

On behalf of the journal, I congratulate you and your co-authors on this achievement. You will be contacted shortly by our production team with the next steps regarding copyediting, proof review, and final publication.

Thank you again for choosing PeerJ as the venue for your work.

Sincerely,
Dr. Gustavo Rodrigues Pedrino
Academic Editor
PeerJ

External reviews were received for this submission. These reviews were used by the Editor when they made their decision, and can be downloaded below.